# Network-to-Network Translation with Conditional Invertible Neural Networks

**Robin Rombach**\*  **Patrick Esser**\*  **Björn Ommer**
IWR, HCI, Heidelberg University
`firstname.lastname@iwr.uni-heidelberg.de`

## Abstract

Given the ever-increasing computational costs of modern machine learning models, we need to find new ways to reuse such expert models and thus tap into the resources that have been invested in their creation. Recent work suggests that the power of these massive models is captured by the representations they learn. Therefore, we seek a model that can relate between different existing representations and propose to solve this task with a conditionally invertible network. This network demonstrates its capability by (i) providing generic transfer between diverse domains, (ii) enabling controlled content synthesis by allowing modification in other domains, and (iii) facilitating diagnosis of existing representations by translating them into interpretable domains such as images. Our domain transfer network can translate between fixed representations without having to learn or finetune them. This allows users to utilize various existing domain-specific expert models from the literature that had been trained with extensive computational resources. Experiments on diverse conditional image synthesis tasks, competitive image modification results and experiments on image-to-image and text-to-image generation demonstrate the generic applicability of our approach. For example, we translate between BERT and BigGAN, state-of-the-art text and image models to provide text-to-image generation, which neither of both experts can perform on their own.

## 1 Introduction

One of the key features of intelligence is the ability to combine and transfer information between diverse domains and modalities [12, 73, 65, 68, 61]. In contrast, artificial intelligence research has made great progress in learning powerful representations for *individual* domains [28, 71, 19, 69, 15, 5] that can even achieve superhuman performance on confined tasks such as traffic sign recognition [10, 11], image classification [29] or question answering [15]. However, learning representations for different domains that also allow a domain-to-domain transfer of information between them is significantly more challenging [2]: There is a trade-off between the expressiveness of individual domain representations and their compatibility to another to support transfer. While for limited training data multimodal learning has successfully trained representations for different domains together [66, 74], the overall most powerful domain-specific representations typically result from training huge models specifically for *individual* challenging domains using massive amounts of training data and computational resources, *e.g.* [19, 69, 5]. With the dawn of even more massive models like the recently introduced GPT-3 [5], where training on only a single domain already demands most of the available resources, we must find new, creative ways to make use of these powerful models, which none but the largest institutions can afford to train and experiment with, and thereby utilize the huge amount of resources and knowledge which are distilled into the model's representations—in other words, we have to find ways to cope with "The Bitter Lesson" [67].

---

Code available at `https://github.com/CompVis/net2net`.

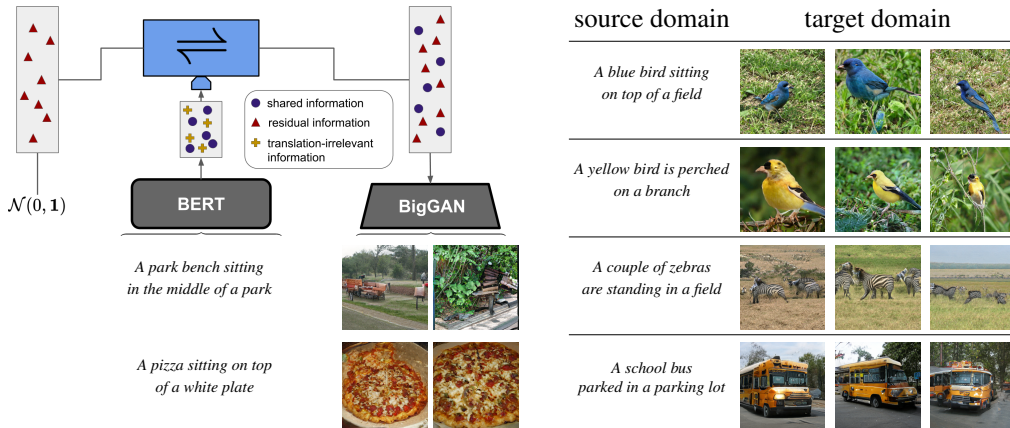

Figure 1: *BERT* [15] to *BigGAN* [4] transfer: Our approach enables translation between fixed off-the-shelve expert models such as BERT and BigGAN without having to modify or finetune them.

Consequently, we seek a model for generic domain-to-domain transfer between arbitrary fixed representations that come from highly complex, off-the-shelf, state-of-the-art models and we learn a domain translation that does not alter or retrain the individual representations but retains the full capabilities of original expert models. This stands in contrast to current influential domain transfer approaches [41, 51, 83] that require learning or finetuning existing domain representations to facilitate transfer between them.

Since different domains are typically not isomorphic to another, *i.e.* translations between them are not uniquely determined, the domain translation between fixed domain representations requires learning the corresponding ambiguities. For example, there are many images which correspond to the same textual description and vice versa. To faithfully translate between domains, we employ a conditional invertible neural network (cINN) that also explicitly captures these transfer uncertainties. The INN conditionally learns a unique translation of one domain representation together with its complementary residual onto another. This generic network-to-network translation between arbitrary models can efficiently transfer between diverse state-of-the-art models such as transformer-based natural language model BERT [15] and a BigGAN [4] for image synthesis to achieve competitive text-to-image translation, see Fig. 1.

To summarize, our contributions are as follows: We (i) provide a generic approach that allows to translate between fixed off-the-shelf model representations, (ii) learns the inherent ambiguity of the domain translation, which facilitates content creation and model diagnostics, and (iii) enables compelling performance on various different domain transfer problems. We (iv) make transfer between domains and datasets computationally affordable, since our method does not require any gradient computations on the expert models but can directly utilize existing representations.

## 2   Related Work

The majority of approaches for deep-learning-based domain-to-domain translation are based on generative models and therefore rely on Variational Autoencoders (VAEs) [37, 58], Generative Adversarial Networks (GANs) [27], autoregressive models [70], or normalizing flows [50] obtained with invertible neural networks (INNs) [16, 17]. Generative models transform samples from a simple base distribution, mainly a standard normal or a uniform distribution, to a complex target distribution, *e.g.* the distribution of (a subset of) natural images.

Sampling the base distribution then leads to the generation of novel content. Recent works [76, 22] also utilize INNs to transform the latent distribution of an autoencoder to the base distribution. A simple structure of the base distribution allows rudimentary control over the generative process in the form of vector arithmetic applied to samples [53, 56, 62, 26], but more generally, providing control over the generated content is formulated as conditional image synthesis. In its most basic form, conditional image synthesis is achieved by generative models which, in addition to a sample from the base distribution, take class labels [47, 36] or attributes [30] into account. More complex conditioning information are considered in [82, 55], where textual descriptions provide more fine-grained control

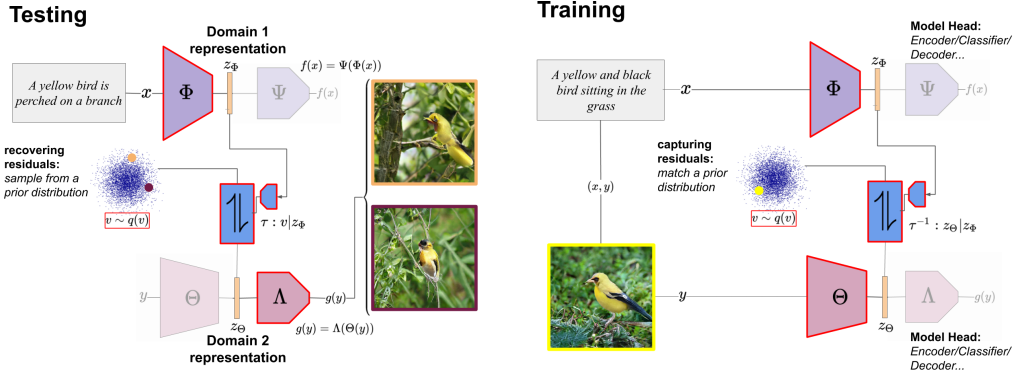

Figure 2: Proposed architecture. We provide post-hoc model fusion for two given deep networks $f = \Phi \circ \Psi$ and $g = \Theta \circ \Lambda$ which live on arbitrary domains $\mathcal{D}_x$ and $\mathcal{D}_y$. For deep representations $z_\Phi = \Phi(x)$ and $z_\Theta = \Theta(y)$, a conditional INN $\tau$ learns to transfer between them by modelling the ambiguities w.r.t. the translation as an explicit residual, enabling transfer between given off-the-shelf models and their respective domains.

over the generative process. A wide range of approaches can be characterized as image-to-image translations where both the generated content and the conditioning information is given by images. Examples for conditioning images include grayscale images [84], low resolution images [38], edge images [33], segmentation maps [51, 7] or heatmaps of keypoints [23, 43]. Many of these approaches build upon [33], which introduced a unified approach for image-to-image translation. We take this unification one step further and provide an approach for a wide range of conditional content creation, including class labels, attributes, text and images as conditioning. In the case of image conditioning, our approach can be trained either with aligned image pairs as in [33, 7, 51] or with unaligned image pairs as in [87, 39, 9, 32, 21]. While many works on generative models focus on relatively simple datasets containing little variations, *e.g.* CelebA [42] containing only aligned images of faces, [4, 19] demonstrated the possibility to apply these models to large-scale datasets such as ImageNet [14]. However, such experiments require a computational effort which is typically far out of reach for individuals. Moreover, the need to retrain large models for experimentation hinders rapid prototyping of new ideas and thus slows down progress. Making use of pre-trained neural networks can significantly reduce the computational budget and training time. For discriminative tasks, the ability to effectively reuse pre-trained neural networks has long been recognized [54, 18, 80]. For generative tasks, however, there are less works that aim to reuse pre-trained networks efficiently. Features obtained from pre-trained classifier networks are used to derive style and content losses for style transfer algorithms [24], and they have been demonstrated to measure perceptual similarity between images significantly better than pixelwise distances [44, 85]. [81, 45] find images which maximally activate neurons of pre-trained networks and [60] shows that improved synthesis results are obtained with adversarially robust classifiers. Instead of directly searching over images, [49] uses a pre-trained generator network of [20], where it was used to reconstruct images from feature representations. However, these approaches are limited to neuron activation problems, rely on per-example optimization problems, which makes synthesis slow, and do not take into account the probabilistic nature of the conditional synthesis task, where a single conditioning corresponds to multiple outputs. To address this, [48] learns an autoregressive model conditioned on specific layers of pre-trained models and [63] a GAN based decoder conditioned on a feature pyramid of a pre-trained classifier. In contrast, our approach efficiently utilizes pre-trained models, both for conditioning as well as for image-synthesis, such that their combination provides new generative capabilities for content creation through conditional sampling, without requiring the pre-trained models to be aware of these emerging capabilities.

# 3   Approach

Our goal is to learn relationships and transfer between representations of different domains obtained from off-the-shelf models, see Fig. 2. To be generally applicable to complex state-of-the-art representations, we only assume the availability of already trained models, but no practical access to their training procedure due to their complexity [15] or missing components (*e.g.* a discriminator, which was not released for [19]). Let $\mathcal{D}_x$ and $\mathcal{D}_y$ be two domains we want to transfer between. Moreover,

$f(x)$ denotes an expert model that has been trained to map $x \in \mathcal{D}_x$ onto desired outputs, *e.g.* class labels in case of classification tasks, or synthesized images for generative image models. To solve its task, a neural network $f$ has learned a latent representation $z_\Phi = \Phi(x)$ of domain $\mathcal{D}_x$ in some intermediate layer, so that subsequent layers $\Psi$ can then solve the task as $f(x) = \Psi(\Phi(x))$. For $y \in \mathcal{D}_y$ let $g(y) = \Lambda(\Theta(y))$ be another, totally different model that provides a feature representation vector $z_\Theta = \Theta(y)$.

In general, we cannot expect a translation from $x$ to $y$ to be unique, since two arbitrary domains and their representations are not necessarily isomorphic. For example, a textual description $x$ of an image $y$ usually leaves many details open and the same holds in the opposite direction, since many different textual descriptions are conceivable for the same image by focusing on different aspects. This implies a non-unique mapping from $z_\Phi$ to $z_\Theta$. Moreover, much of the power of model $f$ trained for a specific task stems from its ability to ignore task-irrelevant properties of $x$. The invariances of $z_\Phi$ with respect to $z_\Theta$ further increase the ambiguity of the domain translation. Obtaining a plausible $z_\Theta$ for a given $z_\Phi$ is therefore best described probabilistically as sampling from $p(z_\Theta|z_\Phi)$. Our goal is to model this process with a translation function $\tau$. Thus, we must introduce a residual $v$, such that for a given $z_\Phi$, $v$ uniquely determines $z_\Theta$ resulting in the translation function $\tau$:

$$z_\Theta = \tau(v|z_\Phi) \tag{1}$$

**Learning a Domain Translation $\tau$:** How can we estimate $v$? $v$ must capture all information of $z_\Theta$ not represented in $z_\Phi$, but no information that is already represented in $z_\Phi$. Hence, to infer $v$, we must take into account both $z_\Theta$, to extract information, and $z_\Phi$, to discard information. The unique determination of $z_\Theta$ from $v$ for a given $z_\Phi$ implies the existence of the inverse of $\tau$, when considered as a function of $v$. Thus for every $z_\Phi$, the inverse $\tau^{-1}(\cdot|z_\Phi)$ of $\tau(\cdot|z_\Phi)$ exists,

$$v = \tau^{-1}(z_\Theta|z_\Phi). \tag{2}$$

This structure of $\tau$ is most naturally represented by a conditionally invertible neural network (cINN), for which $\tau^{-1}$ can be explicitly computed, and which we build from affine coupling [17], actnorm [35] and shuffling layers, see Sec. G.1. It then remains to derive a learning task which ensures that information of $z_\Phi$ is discarded in $v$. To formalize this goal, we consider training pairs $\{(x, y)\} \subset \mathcal{D}_x \times \mathcal{D}_y$ and their corresponding features $\{(z_\Phi, z_\Theta)\}$ as samples from their joint distribution $p(z_\Phi, z_\Theta)$. $v$ can then be considered as a random variable via the process

$$v = \tau^{-1}(z_\Theta|z_\Phi), \quad \text{with } z_\Phi, z_\Theta \sim p(z_\Phi, z_\Theta). \tag{3}$$

Then $v$ discards all information of $z_\Phi$ if $v$ and $z_\Phi$ are independent. To achieve this independence, we minimize the distance between the distribution $p(v|z_\Phi)$ induced by $\tau$ via Eq. (3) and some prior distribution $q(v)$. The latter can be chosen arbitrarily as long as it is independent of $z_\Phi$, its density can be evaluated and samples can be drawn. In practice we use a standard normal distribution. Using the invertibility of $\tau$, we can then explicitly calculate the Kullback-Leibler divergence between $p(v|z_\Phi)$ and $q(v)$ averaged over $z_\Phi$ (see Sec. B for the derivation):

$$\mathbb{E}_{z_\Phi} \mathrm{KL}(p(v|z_\Phi)|q(v)) = \mathbb{E}_{z_\Theta, z_\Phi} \left\{ -\log q(\tau^{-1}(z_\Theta|z_\Phi)) - |\det J_{\tau^{-1}}(z_\Theta|z_\Phi)| \right\} - H(z_\Theta|z_\Phi). \tag{4}$$

Here, $\det J_{\tau^{-1}}$ denotes the determinant of the Jacobian of $\tau^{-1}$ and $H$ is the (constant) data entropy. If $\tau$ minimizes Eq. (4), we have $p(v|z_\Phi) = q(v)$, such that the desired independence is achieved. Moreover, we can now simply achieve the original goal of sampling from $p(z_\Theta|z_\Phi)$ by translating from $z_\Phi$ to $z_\Theta = \tau(v|z_\Phi)$ with $v$ sampled from $q(v)$, which properly models the inherent ambiguity.

**Interpretation as Information Bottleneck:** One of the main goals of minimizing Eq. (4) is the independence of $v$ and $z_\Phi$. While it is clear that this independence is achieved by a minimizer, Eq. (4) is also an upper bound on the mutual information $I(v, z_\Phi)$ between $v$ and $z_\Phi$. Thus, its minimization works directly towards the goal of independence. Indeed, following [1], we have

$$I(v, z_\Phi) = \int_{v, z_\Phi} p(v, z_\Phi) \log \frac{p(v, z_\Phi)}{p(v)p(z_\Phi)} = \int_{v, z_\Phi} p(v, z_\Phi) \log p(v|z_\Phi) - \int_v p(v) \log p(v) \tag{5}$$

Positivity of the KL divergence implies $\int p(v) \log p(v) \geq \int p(v) \log q(v)$, such that

$$I(v, z_\Phi) \leq \int_{v, z_\Phi} p(v, z_\Phi) \log \frac{p(v|z_\Phi)}{q(v)} = \mathbb{E}_{z_\Phi} \mathrm{KL}(p(v|z_\Phi)|q(v)) = \text{Eq. (4)} \tag{6}$$

In contrast to the deep variational information bottleneck [1], our use of a cINN has the advantage that it does not require the hyperparameter $\beta$ to balance the independence of $v$ and $z_\Phi$ against their ability to reconstruct $z_\Theta$. The cINN guarantees perfect reconstruction abilities of $z_\Theta$ due to its invertible architecture and it thus suffices to minimize $I(v, z_\Phi)$ on its own.

**Domain Transfer Between Fixed Models:**   At inference time, we obtain translated samples $z_\Theta$ for given $z_\Phi$ by sampling from the residual space $v$ given $z_\Phi$ and then applying $\tau$,

$$z_\Theta \sim p(z_\Theta|z_\Phi) \quad \Longleftrightarrow \quad v \sim q(v),\ z_\Theta = \tau(v|z_\Phi). \tag{7}$$

After training our domain translator, transfer between $\mathcal{D}_x$ and $\mathcal{D}_y$ is thus achieved by the following steps: (i) sample $x$ from $p(x)$, (ii) encode $x$ into the latent space $z_\Phi = \Phi(x)$ of expert model $f$, (iii) sample a residual $v$ from the prior $q(v)$, (iv) conditionally transform $z_\Theta = \tau(v|z_\Phi)$, and (v) decode $z_\Theta$ into the domain $\mathcal{D}_y$ of the second expert model: $y = \Lambda(z_\Theta)$.

Note that this approach has multiple advantages: (i) hidden representations usually have lower dimensionality than $x$, which makes transfer between arbitrary complex domains affordable, (ii) the cINN $\tau$ can be trained by minimizing the negative log-likelihood, independent of the domains $\mathcal{D}_x$ and $\mathcal{D}_y$, and (iii) the approach does not require to take any gradients w.r.t. the expert models $f$ and $g$.

## 4   Experiments

We investigate the wide applicability of our approach by performing experiments with multiple domains, datasets and models: (1) text-to-image translation by combination of BigGAN and BERT, (2) exploration of the reusability of a fixed autoencoder combined with multiple models including a ResNet-50 classifier and a DeepLabV2 [6] segmentation model for various image-to-image translation tasks and diagnostic insights into the respective models, and (3) comparison to existing methods for image modification and applications in exemplar-guided and unpaired image translation tasks. As our method does not require gradients w.r.t. the models $f$ and $g$, training of the cINN can be conducted on a single Titan X GPU.

### 4.1   Translation to BigGAN

This section is dedicated to the task of using a popular but computationally expensive to train expert model as an image generator: BigGAN [4], achieving state-of-the-art FID scores [31] on the ImageNet dataset. As most GAN frameworks in general and BigGAN in particular do not include an explicit encoder into a latent space, we aim to provide an encoding from an arbitrary domain by using an appropriate expert model $f$. Aiming at the reusability of a fixed BigGAN $g$, and given the hidden representation $z_\Phi = \Phi(x)$ of the expert model $f = \Psi \circ \Phi$, we want to find a mapping between $z_\Phi$ and the latent space $z_\Theta$ of BigGAN's generator $\Lambda$, where, in accordance with Fig. 2, $\Theta \equiv \mathbb{1}$ and $g = \Lambda$. Technical details regarding the training of our cINN can be found in Sec. G.2.

**BERT-to-BigGAN Translation:**   The emergence of transformer-based networks [72] has led to an immense leap in the field of natural language processing, where a popular model is the so-called BERT model. Here, we make use of a variant of the original model, which modifies BERT such that it produces a latent space in which input sentences can be compared for similarity via the cosine-distance measure [57]. We aim to combine this representational power with the synthesis capabilities of BigGAN and thus train our model $\tau$ to map from the language representations $z_\Phi = \Phi(x)$ into the latent space $z_\Theta$ of BigGAN's generator as described above; hence $f = \Phi$ and $\Psi = \mathbb{1}$. During training, access to textual descriptions is obtained by using a captioning model as in [77], trained on the COCO [40] dataset. In a nutshell, at training time, we sample $z_\Theta$, produce a corresponding image $\Lambda(z_\Theta)$, utilize [77] to produce a text-caption $x$ describing the image and subsequently produce a sentence representation $z_\Phi = \Phi(x)$ which we use to minimize the overall objective Eq. (4). Results

Table 1: Inception and FID scores for BERT-to-BigGAN transfer on captions from COCO-stuff. Our approach is on-par with the current state of the art but does not require training of a text-encoder and image-decoder.

|  | our | SD-GAN [79] | AttnGAN [78] | StackGAN [82] | DM-GAN [88] | MirrorGAN [52] | HDGAN [86] |
|---|---|---|---|---|---|---|---|
| IS ↑ | **34.7 ± 0.3** | **35.7 ± 0.5** | 25.9 ± 0.5 | 8.5 ± 0.1 | 30.5 ± 0.6 | 26.5 ± 0.4 | 11.9 ± 0.2 |
| FID ↓ | **30.63** | - | 35.49 | - | 32.64 | - | - |

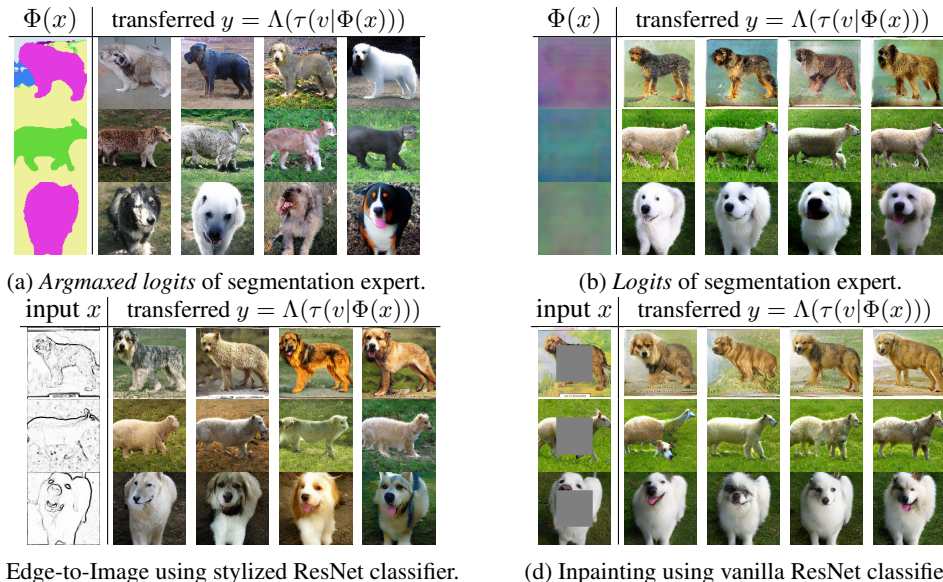

$\Phi(x)$ | transferred $y = \Lambda(\tau(v|\Phi(x)))$

(a) *Argmaxed logits* of segmentation expert.

$\Phi(x)$ | transferred $y = \Lambda(\tau(v|\Phi(x)))$

(b) *Logits* of segmentation expert.

input $x$ | transferred $y = \Lambda(\tau(v|\Phi(x)))$

(c) Edge-to-Image using stylized ResNet classifier.

input $x$ | transferred $y = \Lambda(\tau(v|\Phi(x)))$

(d) Inpainting using vanilla ResNet classifier.

Figure 3: Different Image-to-Image translation tasks solved with a single AE $g$ and different experts $f$.

can be found in Fig. 1 and Tab. 1. Our model captures both fine-grained and coarse descriptions (*e.g.* blue bird vs. yellow bird; school bus vs. pizza) and is able to synthesize images with highly different content, based on given textual inputs $x$. Although not being trained on the COCO images, Tab. 1 shows that our model is highly competitive and on-par with the state-of-the art in terms of Inception [59] and FID [31] scores where available.

## 4.2   Repurposing a single target generator for different source domain models

Here, we train the cINN $\tau$ conditioned on hidden representations of networks such as classifiers and segmentation models, and thereby show that standard classifiers on arbitrary source domains can drive the same generator to create content by transfer. Refering to Fig. 2, this means that $f$ is represented by a classifier/segmentation model, whereas $\Lambda$ is a decoder of an autoencoder that is pretrained on a dataset of interest. Furthermore, we evaluate the ability of our approach to combine a single, powerful domain expert (the autoencoder) with different source models to solve a variety of image-to-image translation tasks. The autoencoder is trained on a combination of all carnivorous animal classes in ImageNet and images of the *AwA2* dataset [75], split into 211306 training images and 10000 testing images, which we call the *Animals* dataset. The details regarding architecture and training of this autoencoder are provided in Sec. F.

**Image-to-Image Translation:**   In Fig. 3, we investigate the translation from different source domain models $\Phi$ onto the same generator $\Lambda$ using our cINN $\tau$. In Fig. 3a, $f$ is a segmentation network trained on COCOStuff, and $\Phi = f$, *i.e.* $z_\Phi$ is given by the final segmentation output of the network. This case corresponds to a translation from segmentation masks to images and we observe that our approach can successfully fuse the segmentation model with the autoencoder to obtain a wide variety of generated image samples corresponding to a given segmentation mask. Fig. 3b uses the same segmentation network for $f$, but this time, $\Phi$ are the logit predictions of the network (visualized by a projection to RGB values). The diversity of generated samples is greatly reduced compared to Fig. 3a, which indicates that logits still contain a lot of information which are not strictly required for segmentation, *e.g.* the color of animals. This shows how different layers of an expert can be selected to obtain more control over the synthesis process.

In Fig. 3c, we consider the task of translating edge images to natural images. Here, $x$ is obtained through the Sobel filter, and we choose a ResNet pretrained for image classification on stylized ImageNet as a domain expert for edge images, as it has shown sensitivity to shapes [25]. This combination of $\Phi$ and $\Lambda$ through $\tau$ enables edge-to-image translation. Fig. 3d shows an image inpainting task, where $x$ is a masked image. In this case, large portions of the shape are missing from the image but the unmasked regions contain texture patches. This makes a ResNet pretrained

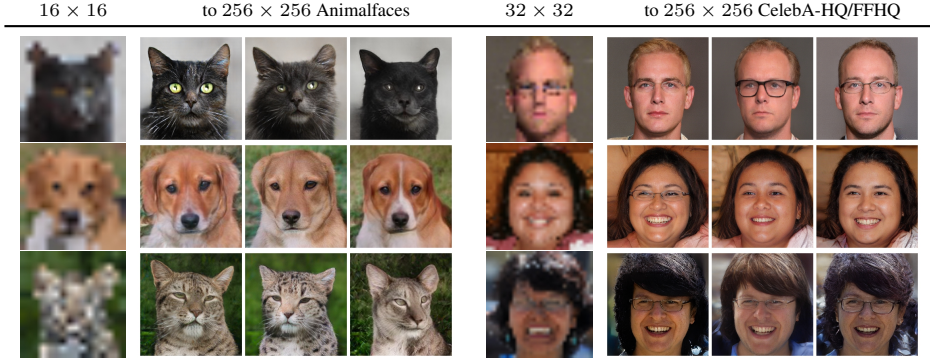

| $16 \times 16$ | to $256 \times 256$ Animalfaces | $32 \times 32$ | to $256 \times 256$ CelebA-HQ/FFHQ |

Figure 4: Superresolution with Network-to-Network Translation. Here, we use our cINN to combine two autoencoders $f$ and $g$ to generatively combine two autoencoders living on image scales $32 \times 32$ and $256 \times 256$.

for image classification on ImageNet a suitable domain expert due to its texture bias. The samples demonstrate that textures are indeed faithfully preserved.

Furthermore, we can employ the same approach for generative superresolution. Fig. 4 shows the resulting transfer when using our method for combining two autoencoders, which are trained on different scales. More precisely, $f$ is an autoencoder trained on images of size $32 \times 32$, while $g$ is an autoencoder of $256 \times 256$ images. The samples show that the model captures the ambiguities w.r.t. this translation and thereby enables efficient superresolution.

**Model Diagnosis:**   Besides being applicable for content creation, our approach to recovering the invariances of $f$ can also serve as a diagnostic tool for model interpretation. By comparing the generated samples $y = \Lambda(z_\Theta)$ (see Eq. (7)) conditioned on representations $z_\Phi = \Phi(x)$ extracted from *different layers* of $f$, we see how the invariances increase with increasing layer depth and can thereby visualize what the model has learned to ignore. Using the same segmentation model $f$ and autoencoder $g$ as described above, we visualize the invariances and model representations for different layers on the *Animals* and a web-scraped *Landscapes* dataset. For the latter, Fig. 5 demonstrates how the model $f$ has learned to discard information in later layers; *e.g.* the variance in the synthesized outputs increases (such as different lightings or colors for the same scene). The corresponding experiment on the *Animals* dataset is presented in Sec. E. There, we also study the importance of faithfully modeling ambiguities of the translation by replacing the cINN with an MLP, which in contrast to the cINN fails to translate deep representations.

Note that all results in Fig. 3 and Fig. 5, 7a were obtained by combining a single, generic autoencoder $g$, which has no capabilities to process inputs of $\mathcal{D}_x$ on its own, and different domain experts $f$, which possess no generative capabilities at all. These results demonstrate the feasibility of solving a wide-range of image-to-image tasks through the fusion of pre-existing, task-agnostic experts on image domains $\mathcal{D}_x, \mathcal{D}_y$. Moreover, choosing different layers of the expert $f$ provides additional, fine-grained control over the generation process.

| $x$ | early layer: $\Lambda(z_\Theta)$ | middle layer: $\Lambda(z_\Theta)$ | last layer of $f$: $\Lambda(z_\Theta)$ |

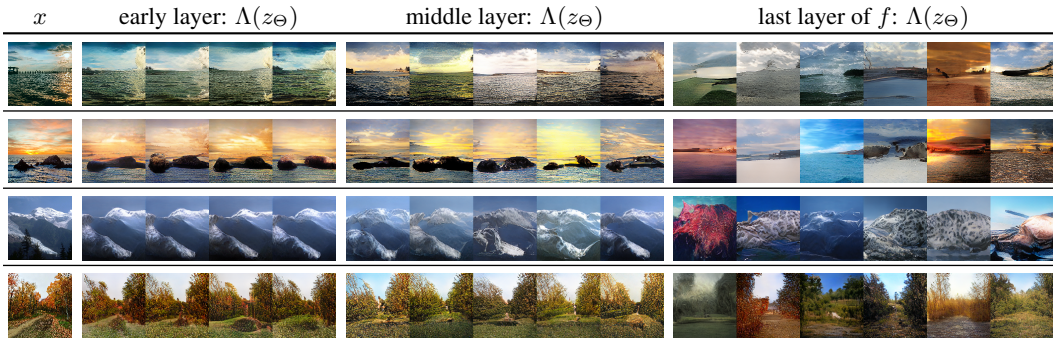

Figure 5: Translating different layers of an expert model $f$ to the representation of an autoencoder $g$ reveals the learned invariances of $f$ and thus provides diagnostic insights. Here, $f$ is a segmentation model, while $g$ is the same AE as in Sec. 4.2. For $z_\Phi = \Phi(x)$, obtained from different layers of $f$, we sample $z_\Theta$ as in Eq. (7) and synthesize corresponding images $\Lambda(z_\Theta)$.

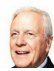                           

| input | method | *hair* | *glasses* | *gender* | | input | method | *beard* | *age* | *smiling* |
|---|---|---|---|---|---|---|---|---|---|---|
| FID | our | **15.18** | **37.32** | **16.38** | | FID | our | **12.02** | **10.77** | **9.57** |
| | [8] | 20.94 | 41.27 | 20.04 | | | [8] | 19.88 | 21.77 | 14.47 |

Figure 6: We directly consider attribute vectors for $z_\Phi$ to perform attribute modifications. We show both qualitative comparisons to [8], obtained by changing a single attribute of the input, as well as quantiative comparisons of FID scores, obtained after flipping a single attribute for all images of the test set. The results demonstrate the value of reusing a powerful, generic autoencoder (AE) $g$ and repurposing it via our approach for a specific task, such as attribute modification, instead of learning an AE and the modification task simultaneously.

## 4.3 Evaluating image modification capabilities of our generic approach

**Attribute Modification:** To compare our generic approach against task-specific approaches, we compare its ability for attribute modification on face images to those of [8]. We train the same autoencoder $g$ as in the previous section on CelebA [42], and directly use attribute vectors as $z_\Phi$. For an input image $y$ with attributes $z_\Phi$, we synthesize versions with modified attributes $z_\Phi^*$. In each column of Fig. 6, we flip the binary entry of the corresponding attribute to obtain $z_\Phi^*$. To obtain the modified image, we first compute $z_\Theta = \Theta(y)$ and use its corresponding attribute vector $z_\Phi$ to obtain its attribute invariant representation $v = \tau^{-1}(z_\Theta | z_\Phi)$. We then mix it again with the modified attribute vector to obtain $z_\Theta^* = \tau(v | z_\Phi^*)$, which can be readily decoded to the modified image $y^* = \Lambda(z_\Theta^*)$.

Qualitative results in Fig. 6 demonstrate successful modification of attributes. In comparison to [8], our approach produces more coherent changes, *e.g.* changing gender causes changes in hair length and changes in the beard attribute have no effect on female faces. This demonstrates the advantage of fusing attribute information on a low-dimensional representation of a generic autoencoder. Overall, our approach produces images of higher quality, as demonstrated by the FID scores [31] in Fig. 6. Note that FID-scores are calculated w.r.t. the complete dataset, explaining the high FID scores for attribute *glasses*, where images consistently possess a large black area.

**Exemplar-Guided Translation:** Another common image modification task is exemplar-guided image-to-image translation [51], where the semantic content and spatial location is determined via a label map and the style via an exemplar image. To approach this task, we utilize the same segmentation model and autoencoder as in Sec. 4.2. As before, we use the last layer of the segmentation model to represent semantic content and location. For a given exemplar $y$, we can then extract its residual $v = \tau^{-1}(\Theta(y) | \Phi(y))$ and the segmentation representation $z_\Phi = \Phi(x)$ of another image $x$. Due to the independence of $v$ and $z_\Phi$, we can now transfer $z_\Phi$ under the guidance of $v$ to a recombined $z_\Theta^* = \tau(v | z_\Phi)$ which is readily decoded to an image $y^* = \Lambda(z_\Theta^*)$ as shown in Fig. 7a.

**Unsupervised Disentangling of Shape and Appearance:** Our approach also allows for an unsupervised variant of the previous task, as shown in Fig. 7b. Here, we use a random spatial deformation $d$ (see Sec. C) to define $\Phi = \Theta \circ d$. Thus, $z_\Phi$ and $z_\Theta$ share the same appearance but differ in pose, which is distilled into $v$. For exemplar guided synthesis, the roles of $x$ and $y$ are now swapped.

**Unpaired Image Translation:** Fig. 8 demonstrates results of our approach applied to unpaired image-to-image translation. Here, we use the same setup as for attribute modification, but train various cINN models for unpaired transfer between the following datasets: CelebA and AnimalFaces-HQ [9], FFHQ and CelebA-HQ [34], Anime [3] to CelebA-HQ/FFHQ and Oil Portraits to CelebA-HQ/FFHQ. Details regarding the training procedure can be found in the Sec. D.

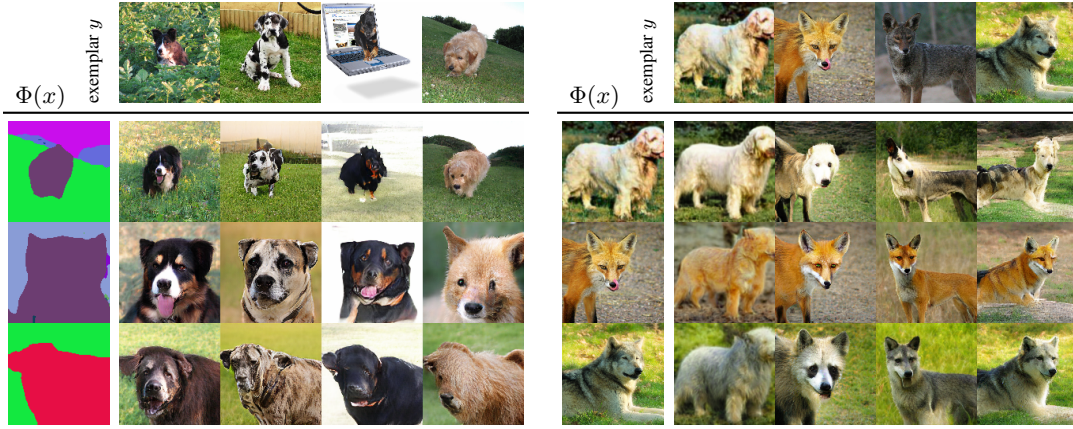

(a) Exemplar-guided image-to-image translation.　　　　(b) Unsupervised shape and appearance disentangling.

Figure 7: In (a), a segmentation representation $\Phi(x)$ is translated under the guidance of the residual $v = \tau^{-1}(\Theta(y)|\Phi(y))$ obtained from exemplar $y$. In (b), $\Phi$ is the same as $\Theta$, but applied after a spatial deformation of its input such that $\tau$ learns to extract a shape representation into $v$, which then controls the target shape.

## 5  Conclusion

This paper has addressed the problem of generic domain transfer between arbitrary fixed off-the-shelf domain models. We have proposed a conditionally invertible domain translation network that faithfully transfers between existing domain representations without altering them. Consequently, there is no need for costly or potentially even infeasible retraining or finetuning of existing domain representations. The approach is *(i) flexible:* Our cINN for translation as well as its optimization procedure are independent from the individual domains and, thus, provide plug-and-play capabilities by allowing to plug in arbitrary existing domain representations; *(ii) powerful:* Enabling the use of pretrained expert domain representations outsources the domain specific learning task to these models. Our model can thus focus on the translation alone which leads to improvements over previous approaches; *(iii) convenient and affordable*: Users can now utilize powerful, pretrained models such as BERT and BigGAN for new tasks they were not designed for, with just a single GPU instead of the vast multi-GPU resources required for training such domain models. Future applications include transfer between diverse domains such as speech, music or brain signals.

| Oil-Portrait to Photography | Anime to Photography | FFHQ to CelebA-HQ | FFHQ to AFHQ |
| --- | --- | --- | --- |

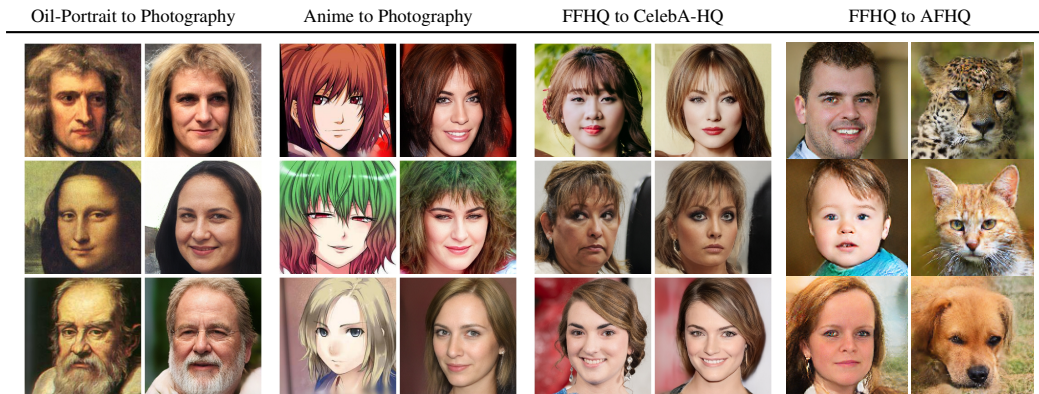

Figure 8: Unpaired Transfer: Bringing oil portraits and animes to live by projecting them onto the FFHQ dataset (column 1 and 2, respectively). Column 3 visualizes the more subtle differences introduced when translating between different datasets of human faces such as FFHQ and CelebA-HQ. Column 4 shows a translation between the more diverse modalities of human and animal faces. See also Sec. 4.3 and D.

## Broader Impact

**Environmental and Economic Aspects:**
Single training runs of large scale models have a large environmental footprint due to the massive computational requirements. It is therefore unreasonable to repeat this effort for every new application. Instead we allow to reuse powerful models, leading to significant reductions in computational demands.

**Boosting serendipity:**
New (scientific) knowledge arises where seemingly unrelated entities are brought together to study their relationship (cf. the 'double projection' proposed by Heinrich Wölfflin a century ago for art history and other image disciplines to easily contextualize diverse imagery of different cultures). Allowing to efficiently connect expert models for diverse data and problems, thus promises to provide the basis for new directions of future research.

**Interdisciplinary research:**
One of the main stumbling blocks for interdisciplinary research (e.g. vision and language) is to bring together expert models for widely different domains. State-of-the-art models are typically developed in and for the individual disciplines. Being able to efficiently combine these disciplinary expert models to solve interdisciplinary problems promises to be an enabling factor for more effective cross-disciplinary research.

**Social equality:**
Training state-of-the art models is typically so costly that only wealthy institutions can afford their transfer and application to different input domains or other subsequent research that would require to retrain them. Computationally efficient transfer of existing models with no need for costly retraining therefore increases the opportunities for economically weaker institutions and countries to have research programs in this field.

**Increasing applicability and impact of research output:**
Large scale research is often funded by public resources and thus there is a responsibility to make results accessible to the public. While pretrained models are often shared publicly, the scope in which they can be applied is significantly widened by our approach.

**Content creation and manipulation:**
Domain transfer applied to controlled image synthesis and modification has wide applicability in the creative industry and beyond. However, it can also be misused for forgery and manipulation.

## Acknowledgments and Disclosure of Funding

This work has been supported in part by the German Research Foundation (DFG) projects 371923335, 421703927, and EXC 2181/1 - 390900948, the German federal ministry BMWi within the project "KI Absicherung" and a hardware donation from NVIDIA Corporation.

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
