[Supplementary Material]

# Network-to-Network Translation with Conditional Invertible Neural Networks

–

## Supplementary Material

## Overview

This appendix provides supplementary material for our work *Network-to-Network Translation with cINNs*. Firstly, in Sec. A, we discuss and compare computational requirements and resulting costs of our model to other approaches, *e.g.* BigGAN. Next, in Sec. B, we provide a derivation of the training objective (see Eq.(4)). Its interpretation as an information bottleneck allows for unsupervised disentangling of shape and appearance, which is demonstrated in Sec. C and Fig. S1. Sec. D then provides additional examples (Fig. S2) and technical details for the unpaired image translation as presented in Fig. 8 (Sec. 4.3). Subsequently, we continue with an ablation study in Sec. E, where we analyze the performance of our approach by replacing the conditional invertible neural network with a multilayer perceptron. Next, Sec. F presents the architecture and training procedure of our reusable autoencoder model $g$, which we used to obtain the results shown in Fig. 3, 5, 7a, 7b, S1, S4, S5 and S6. Finally, Sec. G provides details on (i) the architecture of the cINN and (ii) the translation from BERT to BigGAN, *c.f.* Sec. 4.1.

## A  Computational Cost and Energy Consumption

In Tab. S1 we compare computational costs of our cINN to those of BERT[2], BigGAN[3] and FUNIT.[4] The Table shows that, once strong domain experts are available, they can be repurposed by our approach in a time-, energy- and cost-effective way. With training costs of our cINN being two orders of magnitude smaller than the training costs of the domain experts, the latter are amortized over all the new tasks that can be solved by recombining experts with our approach.

| Model | Time [days] | Hardware | Energy [kWh] | Cost [EUR] | $CO_2$ [kg] |
|---|---|---|---|---|---|
| **our cINN** | $\leq 1$ | 1 NVIDIA Titan X | **14.4** | **3.11** | **4.26** |
| BigGAN [4] | 15 | 8 NVIDIA V100 | 1260.0 | 272.16 | 372.96 |
| FUNIT [41] | 14 | 8 NVIDIA V100 | 1176.0 | 254.02 | 348.10 |
| BERT [15] | 10.3 | 8 NVIDIA V100 | 865.2 | 186.88 | 256.10 |

Table S1: Comparison of computational costs for a single training run of different models. Energy consumption of a Titan X is based on the recommended system power (0.6 kW) by NVIDIA[5], and energy consumption of eight V100 on the power (3.5 kW) of a NVIDIA DGX-1 system[6]. Costs are based on the average price of 0.216 EUR per kWh in the EU[7], and CO2 emissions on the average emissions of 0.296 kg CO2 per kWh in the EU[8].

# B  Training Objective

**Derivation of Eq.** (4): For a given $z_\Phi$, we use a change of variables, $v = \tau^{-1}(z_\Theta|z_\Phi)$, to express the KL divergence with an integral over $z_\Theta$:

$$\text{KL}(p(v|z_\Phi)|q(v)) = \int_v p(v|z_\Phi) \log \frac{p(v|z_\Phi)}{q(v)} \tag{8}$$

$$= \int_{z_\Theta} p(\tau^{-1}(z_\Theta|z_\Phi)|z_\Phi)|\det J_{\tau^{-1}}(z_\Theta|z_\Phi)|\log \frac{p(\tau^{-1}(z_\Theta|z_\Phi)|z_\Phi)}{q(\tau^{-1}(z_\Theta|z_\Phi))} \tag{9}$$

By definition of $v$ in Eq. (3), the invertibility of $\tau$ allows us to express the conditional probability density function $p(v|z_\Phi)$ of $v$ in terms of the conditional probability density function $p(z_\Theta|z_\Phi)$ of $z_\Theta$:

$$p(v|z_\Phi) = p(\tau(v|z_\Phi)|z_\Phi)|\det J_\tau(v|z_\Phi)| \tag{10}$$

For $v = \tau^{-1}(z_\Theta|z_\Phi)$, the inverse function theorem then implies

$$p(\tau^{-1}(z_\Theta|z_\Phi)|z_\Phi) = p(z_\Theta|z_\Phi)|\det J_{\tau^{-1}}(z_\Theta|z_\Phi)|^{-1} \tag{11}$$

Using Eq. (11) in Eq. (9) gives

$$\text{KL}(p(v|z_\Phi)|q(v)) = \int_{z_\Theta} p(z_\Theta|z_\Phi) \log \frac{p(z_\Theta|z_\Phi)}{q(\tau^{-1}(z_\Theta|z_\Phi))|\det J_{\tau^{-1}}(z_\Theta|z_\Phi)|} \tag{12}$$

$$= \mathbb{E}_{z_\Theta} \left\{ -\log q(\tau^{-1}(z_\Theta|z_\Phi)) - \log|\det J_{\tau^{-1}}(z_\Theta|z_\Phi)| + \log p(z_\Theta|z_\Phi) \right\} \tag{13}$$

Taking the expectation over $z_\Phi$ results in Eq. (4).

Figure S1: Unsupervised disentangling of shape and appearance. Training our approach on synthetically deformed images, $\tau$ learns to extract a disentangled shape representation $v$ from $y$, which can be recombined with arbitrary appearances obtained from $x$. See also Sec. C.

## C  Unsupervised Disentangling of Shape and Appearance

Unsupervised disentangling of shape and appearance aims to recombine shape, *i.e.* the underlying spatial structure, from one image with the appearance, *i.e.* the style, of another image. In contrast to Unpaired Image-to-Image translation of Sec. D, training data does not come partitioned into a discrete set of different image domains, and in contrast to Exemplar-Guided Translation of Sec. 4.3, the task assumes that no shape expert, *e.g.* a segmentation model, is available. To handle this setting, we use the encoder $\Theta$ of the autoencoder trained on *Animals* also for $\Phi$, but always apply a spatial deformation to its inputs. Thus, the pairs $(x, y)$ are given by $(d(y), y)$, where $y$ are the original images from the *Animals* dataset and $d$ is a random combination of horizontal flipping, a thin-plate-spline transformation and cropping. The translation task then consists of the translation of deformed encodings $z_\Phi = \Phi(d(y)) := \Theta(d(y))$ to the original encoding $z_\Theta = \Theta(y)$. After training, we apply our translation network to original images without the transformation $d$. For two images $x$ and $y$, we obtain a shape representation of $y$ from its residual $v = \tau^{-1}(\Theta(y)|\Theta(y))$ and recombine it with the appearance of $x$ to obtain $y^* = \Lambda(\tau(v|\Theta(x)))$. The results in Fig. S1 demonstrate that our translation network succesfully learns a shape representation $v$, which is independent of the appearance and can thus be recombined with arbitrary appearances, see Eq. (6). Being able to operate completely unsupervised demonstrates the generality of our approach.

Figure S2: Additional examples for unpaired translation between human and animal faces as in Fig. 8. Our approach naturally provides translations in both directions (see Sec. D). Inputs are randomly choosen test examples from either the human or the animal data and translated to the respective other one.

Figure S3: Additional examples for unpaired translation of Oil Portraits to FFHQ/CelebA-HQ and Anime to FFHQ/CelebA-HQ. Here, we show samples where the *same* $v$ is projected onto the respective dataset.

# D  Unpaired Image Translation

Unpaired Image-to-Image translation considers the case where only unpaired training data is available. Following [87], let $Y^0 = \{y_i^0\}_{i=1}^N$ be a source set, and $Y^1 = \{y_j^1\}_{j=1}^M$ a target set. The goal is then to learn a translation from source set to target set, with no information provided as to which $y_i^0$ matches which $y_j^1$. We formulate this task as a translation from a set indicator $x := z_\Phi \in \{0, 1\}$ to an output $y$, such that for $z_\Phi = 0$, $y$ belongs to the source set $Y^0$, and for $z_\Phi = 1$, $y$ belongs to the target set. Thus, in the case of unpaired image translation, the domains are given by $\mathcal{D}_x = \{0, 1\}$ and $\mathcal{D}_y = Y^0 \cup Y^1$, with training pairs $\{(0, y_i^0) | i = 0, \ldots, N\} \cup \{(1, y_j^1) | j = 0, \ldots, M\}$. Because the residual $v$ is independent of $z_\Phi$, it captures precisely the commonalities between source and target set and therefore establishes meaningful correspondences between them. To translate $y_i^0$, we first obtain its residual $v = \tau^{-1}(\Theta(y_i^0) | 0)$, and then decode it as an element of the target set $y^* = \Lambda(\tau(v|1))$. Note that the autoencoder $g = \Lambda \circ \Theta$ is always trained on the combined domain $\mathcal{D}_y = Y^0 \cup Y^1$, *i.e.* a combination of the two datasets of interest.

Additional examples for unpaired translation as in Fig. 8 can be found in Fig. S2. One can see that the viewpoint of a face is preserved upon translation, which demonstrates that $v$ learns semantic correspondences between the pose of faces from *different* dataset modalities, without any paired data between them. The same holds for the samples in Fig. S3, where the same $v$ is projected onto different domains $\mathcal{D}_x$.

| $x$ | method | early layer: $\Lambda(z_\Theta)$ | middle layer: $\Lambda(z_\Theta)$ | last layer of $f$: $\Lambda(z_\Theta)$ |
|---|---|---|---|---|

| FID | our | $34.0 \pm 0.1$ | $23.4 \pm 0.7$ | $27.6 \pm 0.1$ |
| | MLP | $24.2$ | $25.6$ | $264.0$ |

Figure S4: Model diagnosis compared to a MLP for the translation. Synthesized samples and FID scores demonstrate that a direct translation with a multilayer perceptron (MLP) does not capture the ambiguities of the translation process and can thus only produce a mean image. In contrast, our cINN correctly captures the variability and produces coherent outputs.

# E    Ablation Study: Replacing our cINN with an MLP

To illustrate the importance of the cINN to model ambiguities of the translation, we demonstrate the effect of replacing our cINN with a (deterministic) multilayer perceptron (MLP) in Fig. S4. The MLP consists of two parts: (i) an embedding part as in Tab. S3b and (ii) the architecture of a fully-connected network which was recently used for neural scene rendering [46]. We perform the same model diagnosis experiment as in Fig. 5, but applied to the *Animals* dataset. For early layers, the translation contains almost no ambiguity and can be handled successfully by both the cINN and the MLP. For a deeper layer of the used segmentation model $f$, the translation has moderate ambiguities, as $f$'s invariances w.r.t. to the input increase. This is not accurately reflected by the multilayer perceptron, because it does not model the space of these invariances. Finally, for the last layer of $f$, the MLP predicts the mean over all possible translation outputs which, due to its large ambiguities, does not result in a meaningful translation anymore, whereas our cINN still samples coherent translation outputs. FID scores in Tab. S4 further validate this behavior for the whole test set.

Figure S5: Additional examples for exemplar-guided image-to-image translation as in Fig. 7a.

# F Architecture and Training of the Autoencoder $g$

All experiments in Sec. 4.2, the exemplar-guided image-to-image translation in Sec. 4.3 as well as the additional results in Fig. S5 and Fig. S6 were conducted using the *same* (*i.e.* same weights and architecture) autoencoder $g$, thereby demonstrating how a single model can be re-used for multiple purposes within our framework.

For the autoencoder, we use a ResNet-101 [29] architecture as encoder $\Theta$, and the BigGAN architecture as the decoder $\Lambda$, see Tab. S2. As we do not use class information, we feed the latent code $z_\Theta$ of the encoder into a a fully-connected layer and use its softmax-activated output as a replacement for the one-hot class vector used in BigGAN. The encoder predicts mean $\Theta(y)_\mu$ and diagonal covariance $\Theta(y)_{\sigma^2}$ of a Gaussian distribution and we use the reparameterization trick to obtain samples $z_\Theta = \Theta(y)_\mu + \mathrm{diag}(\Theta(y)_{\sigma^2})\epsilon$ of the latent code, where $\epsilon \sim \mathcal{N}(0, \mathbb{1})$. For the reconstruction loss $\|\cdot\|$, we use a perceptual loss based on features of a pretrained VGG-16 network [64], and, following [13], include a learnable, scalar output variance $\gamma$. Additionally, we use the PatchGAN discriminator $\mathcal{D}$ from [33] for improved image quality. Hence, given the autoencoder loss

$$\mathcal{L}_{VAE}(\Theta, \Lambda, \gamma) = \mathbb{E}_{\substack{y \sim p(y) \\ \epsilon \sim \mathcal{N}(0,\mathbb{1})}} \left[ \frac{1}{\gamma} \|y - \Lambda(\Theta_\mu(y) + \sqrt{\mathrm{diag}(\Theta_{\sigma^2}(y))}\,\epsilon)\| + \log \gamma \right.$$

$$\left. + \mathrm{KL}\left(\mathcal{N}\left(z_\Theta | \Theta_\mu(y), \mathrm{diag}(\Theta_{\sigma^2}(y))\| \mathcal{N}(0,\mathbb{1})\right)\right) \right], \qquad (14)$$

and the GAN-loss

$$\mathcal{L}_{GAN}(g, \mathcal{D}) = \mathbb{E}_{\substack{y \sim p(y) \\ \epsilon \sim \mathcal{N}(0,\mathbb{1})}} \left[ \log \mathcal{D}(y) + \log\left(1 - \Lambda\left(\Theta_\mu(y) + \sqrt{\mathrm{diag}(\Theta_{\sigma^2}(y))}\,\epsilon\right)\right) \right], \qquad (15)$$

$\Phi(x)$  translating $\Phi(x)$ onto target domain of AE $g$ with different samples $v \sim q(v)$

Figure S6: Additional *Landscape* samples, obtained by translation of the argmaxed logits (*i.e.* the segmentation output) of the segmentation model from Sec. 4.2, 4.3 into the space of our autoencoder $g$, see Sec. 4.2, 4.3. The synthesized examples demonstrate that our approach is able to generate diverse and realistic images from a given label map or through a segmentation model.

the total objective for the training of $g = \{\Theta, \Lambda, \gamma\}$ reads:

$$\{\Theta^*, \Lambda^*, \gamma^*\} = \arg \min_{\Theta, \Lambda, \gamma} \max_{\mathcal{D}} \left[ \mathcal{L}_{VAE}(\Theta, \Lambda, \gamma) + \lambda \mathcal{L}_{GAN}(\{\Theta, \Lambda, \gamma\}, \mathcal{D}) \right]. \qquad (16)$$

This is similar to the improved image metric suggested in [20], but in contrast to their work, we use an *adaptive* weight $\lambda$, computed by the ratio of the gradients of the decoder $\Lambda$ w.r.t. its last layer $\Lambda_L$:

$$\lambda = \frac{\|\nabla_{\Lambda_L}(\mathcal{L}_{rec})\|}{\|\nabla_{\Lambda_L}(\mathcal{L}_{GAN})\| + \delta} \qquad (17)$$

where the reconstruction loss $\mathcal{L}_{rec}$ is given as (*c.f.* Eq. (14)):

$$\mathcal{L}_{rec} = \mathbb{E}_{\substack{y \sim p(y) \\ \epsilon \sim \mathcal{N}(0, \mathbb{1})}} \left[ \frac{1}{\gamma} \|y - \Lambda(\Theta_\mu(y) + \sqrt{\mathrm{diag}(\Theta_{\sigma^2}(y))}\ \epsilon)\| + \log \gamma \right], \qquad (18)$$

and a small $\delta$ is added for numerical stability.

| RGB image $x \in \mathbb{R}^{128 \times 128 \times 3}$ |
| :---: |
| Conv down $\to \mathbb{R}^{64 \times 64 \times 64}$ |
| Norm, ReLU, MaxPool $\to \mathbb{R}^{32 \times 32 \times 64}$ |
| $3\times$ BottleNeck $\to \mathbb{R}^{32 \times 32 \times 256}$ |
| $4\times$ BottleNeck down $\to \mathbb{R}^{16 \times 16 \times 512}$ |
| $23\times$ BottleNeck down $\to \mathbb{R}^{8 \times 8 \times 1024}$ |
| $3\times$ BottleNeck down $\to \mathbb{R}^{4 \times 4 \times 2048}$ |
| AvgPool, FC $\mapsto (\mu, \sigma^2) \in \mathbb{R}^{128} \times \mathbb{R}^{128}$ |

(a) Encoder based on *Resnet-101*.

| $\bar{z} \in \mathbb{R}^{128} \sim \mathcal{N}(\mu, \mathrm{diag}(\sigma^2))$ <br> $3\times$ (FC, LReLU) $\to \mathbb{R}^{256}$ <br> FC, Softmax $\to \mathbb{R}^{1000}$ <br> Embed $\mapsto h \in \mathbb{R}^{128}$ |
| :---: |
| FC$(\bar{z}) \to \mathbb{R}^{4 \times 4 \times 16 \cdot 96}$ |
| ResBlock$(\bar{z},h)$ up $\to \mathbb{R}^{8 \times 8 \times 16 \cdot 96}$ |
| ResBlock$(\bar{z},h)$ up $\to \mathbb{R}^{16 \times 16 \times 8 \cdot 96}$ |
| ResBlock$(\bar{z},h)$ up $\to \mathbb{R}^{32 \times 32 \times 4 \cdot 96}$ |
| ResBlock$(\bar{z},h)$ up $\to \mathbb{R}^{64 \times 64 \times 2 \cdot 96}$ |
| Non-Local Block $\to \mathbb{R}^{64 \times 64 \times 2 \cdot 96}$ |
| ResBlock$(\bar{z},h)$ up $\to \mathbb{R}^{64 \times 64 \times 96}$ |
| Norm, ReLU, Conv up $\to \mathbb{R}^{128 \times 128 \times 3}$ |
| Tanh $\mapsto \bar{x} \in \mathbb{R}^{128 \times 128 \times 3}$ |

(b) Decoder based on *BigGAN*.

Table S2: Autoencoder architecture for the *CelebA* and *Animals* datasets at resolution $128 \times 128$.

## G   Implementation Details

### G.1   Architecture of the conditional INN

Figure S7: A single (conditionally) invertible block used to build our cINN. We build the cINN from $n = 20$ of these blocks for all of our experiments.

In our implementation, the conditional invertible neural network (cINN) consists of a sequence of INN-blocks as shown in Fig. S7, and a non-invertible embedding module $H$ which provides the conditioning information. Each INN-block is build from (i) an alternating affine coupling layer [17], (ii) an activation normalization (*actnorm*) layer [35], and (iii) a fixed permutation layer, which effectively mixes the components of the network's input. Given an input $z \in \mathbb{R}^D$ and additional conditioning information $y$, we pre-process the latter with a neural network $H$ as

$$h = H(y), \tag{19}$$

and a single *conditional* affine coupling layer splits $z$ into two parts $z_{1:d}, z_{d+1:D}$ and computes

$$z_{1:d}, z_{d+1:D} = \mathtt{split}(z) \tag{20}$$

$$z' = \mathtt{concat}\Big( z_{1:d}, s_\theta([z_{1:d}; h]) \odot z_{d+1:D} + t_\theta([z_{1:d}; h]) \Big) \tag{21}$$

where $s_\theta$ and $t_\theta$ are implemented as simple feedforward neural networks, which process the concatenated input $[z_{i:j}, h]$ (see Tab. S3a). The *alternating* coupling layer ensures mixing for all components of $z$:

$$z'' = \mathtt{concat}\Big( s_\theta([z'_{d+1:D}; h]) \odot z_{1:d} + t_\theta([z'_{d+1:D}; h]), z'_{d+1:D} \Big). \tag{22}$$

|  |
| --- |
| input $z \in \mathbb{R}^d$ |
| (FC, LReLU) $\to \mathbb{R}^{8\cdot d}$ |
| $2\times$ (FC, LReLU) $\to \mathbb{R}^{8\cdot d}$ |
| (FC, LReLU) $\to \mathbb{R}^d$ |

(a) Basic fully connected architecture.

|  |
| --- |
| input $y \in \mathbb{R}^{c\times h\times w}$ |
| $n\times$ Conv down, ActNorm, LReLU $\to \mathbb{R}^{64\cdot n\times h/2^n \times w/2^n}$ |
| Flatten, FC $\to \mathbb{R}^d$ |

(b) Embedding module $H$, see Eq. (19).

Table S3: *(a)*: Architecture of the subnetworks $s_\theta$ and $t_\theta$ used to build the normalizing flow described in Sec. 3, G.1. Leaky ReLU (LReLU) uses a slope parameter $\alpha = 0.01$. For the cINN from Sec. 4.1, $d = 268$, while for Sec. 4.2 and Sec. 4.3, $d = 128$. *(b)* Architecture of the embedding module $H$, which is used to pre-process arbitrarily sized conditioning information $y$ via $h = H(y)$. Here, $n$ denotes the number of downsampling steps. If the conditioning $y$ does not have spatial dimensionality, we replace the whole network by a simple feedforward-architecture as in Tab. S3a.

We implement the embedding module $H$ as a simple (convolutional) neural network, see Tab. S3b for details regarding its architecture. Note that $H$ does *not* need to be invertible as it soley processes conditioning information. Hence, given some $h = H(y)$, the network $\tau$ is conditionally invertible. Usually, we train with a batch size of 10-25, which requires 4-12 GB VRAM and converges in less than a day.

## G.2  Training Details for Bert-to-BigGAN Translation

| source domain $x$ | target domain $y$ | source domain $x$ | target domain $y$ |
| --- | --- | --- | --- |
| *Two people on a paddle boat in the water* | | *A close up of a plant with broccoli* | |
| *A close up of a clock on a wall* | | *A fighter jet flying through a cloudy sky* | |
| *A black bear is walking through the woods* | | *A glass of wine sitting on a table* | |
| *A man riding skis down a snow covered slope* | | *A wooden chair sitting in front of a chair* | |
| *A group of horses pulling a carriage down a dirt* | | *A car parked in front of a building* | |

Figure S8: *BERT* [15] to *BigGAN* [4] transfer: Additional examples, which demonstrate high diversity in synthesized outputs.

Training our approach to translate between a model's $f$ representation $z_\Phi = \Phi(x)$ and BigGAN's latent space $z_\Theta$ requires to dequantize the discrete class labels $c$ that BigGAN is trained with. To do so, we consider the stacked vector

$$z_\Theta' = [\tilde{z}, Wc],\qquad(23)$$

consisting of $\tilde{z} \sim \mathcal{N}(0, \mathbb{1})$, $\tilde{z} \in \mathbb{R}^{140}$, sampled from a multivariate normal distribution and $c \in \{0, 1\}^K$, a one-hot vector specifying an ImageNet class ($K = 1000$ classes in total). The matrix

$W$, a part of the generator $\Lambda$, maps the one-hot vector $c$ to $h \in \mathbb{R}^{128}$, *i.e.* $h = Wc$, such that $z'_\Theta$ in Eq. (23) corresponds to a synthesized image, given a pretrained generator of BigGAN. However, as $c$ contains discrete labels, we have to avoid collapse of $\tau$ onto a single dimension of $h$ during training. To this end, we pass the vector $h$ through a small, fully connected variational autoencoder (described in Tab. S4) and replace $h$ by its stochastic reconstruction $\hat{h}$, which effectively performs dequantization, such that:

$$z_\Theta = \left[\tilde{z}, \hat{h}\right], \quad \text{with } z_\Theta \in \mathbb{R}^{268}. \tag{24}$$

Training of $\tau$ is then conducted by sampling $z_\Theta$ as in Eq. (24) and minimizing the objective described in Eq. (4), *i.e.* finding a mapping $\tau$ that conditionally maps $z_\Phi$ and the corresponding ambiguities $v \sim \mathcal{N}(0, \mathbb{1})$ to $g$'s representations $z_\Theta$. Additional results obtained when using this approach to conditionally translate BERT's representation $z_\Phi$ into the latent space of BigGAN (see also Sec. 4.1) can be found in Fig. S8.

| Embedding $h \in \mathbb{R}^{128}$ |
|:---:|
| $3\times$ (FC, LReLU) $\to \mathbb{R}^{4096}$ |
| (FC, LReLU) $\to \mathbb{R}^{128}$ |
| $\mu, \sigma^2$: for each: |
| $3\times$ (FC, LReLU) $\to \mathbb{R}^{4096}$ |
| (FC, LReLU) $\to \mathbb{R}^{128}$ |
| $h \in \mathbb{R}^{128} \sim \mathcal{N}(\mu, \text{diag}(\sigma^2))$ |
| $4\times$ (FC, LReLU) $\to \mathbb{R}^{4096}$ |
| (FC, LReLU) $\to \mathbb{R}^{128}$ |

Table S4: Training a cINN on synthetic BigGAN data requires to dequantize the discrete class information which is used as conditioning information for the decoder. To this end, we make use of a variational autoencoder as described in Sec. G.2, which provides a stochastic reconstruction of its input $h$. For Leaky ReLU, we use a slope parameter of $\alpha = 0.01$.

## Footnotes

[2]BERT requirements: `https://ngc.nvidia.com/catalog/resources/nvidia:bert_for_tensorflow/performance`

[3]BigGAN requirements: `https://github.com/ajbrock/BigGAN-PyTorch`

[4]FUNIT requirements: `https://github.com/NVlabs/FUNIT/`

[4]Titan X Specs: `https://www.nvidia.com/en-us/geforce/products/10series/titan-x-pascal/`

[5]DGX-1 Specs: `https://docs.nvidia.com/dgx/dgx1-user-guide/introduction-to-dgx1.html`

[6]Electricity Prices: `https://ec.europa.eu/eurostat/statistics-explained/index.php?title=Electricity_price_statistics`

[7]CO2 Emission Intensity: `https://www.eea.europa.eu/data-and-maps/daviz/co2-emission-intensity-5`