[Reviews · NeurIPS 2020]

Review 1

Summary and Contributions: In this paper, authors propose a task-to-task translation method that allows to solve domain-to-domain translation task (e.g., image-to-image translation, text-to-image synthesis) with existing pretrained expert models. The proposed method does not involve neither finetuning nor modification of the expert models and only assumes that both models first translate the input objects to latent space. The core idea behind the approach is to learn a mapping between the two expert's latent embeddings via cINNs.

Strengths: The idea behind the model is both novel and paradigm-shifting: state-of-art domain-to-domain models are carefully designed for individual set of domains, whereas authors have shown that their approach can be trained with almost arbitrary experts given that the objects they were trained on and the tasks are reasonably similar. Authors provided both theoretical and empirical justifications of their claims and performed exceptionally well-designed analysis of the models behavior. Authors also provided some experimental results with using the embedding of different layers which allows better understanding of the model behavior. Finally, the paper includes impressive qualitative and quantitative results of the method in various tasks, such as image-to-image translation and text-to-image synthesis and show that the proposed method indeed allows efficient domain-to-domain transfer of relevant information while preserving the target expert's ability to generate variable objects.

Weaknesses: The paper is already great. There are only two things that I would improve here: 1. Abbreviation cINNs in the title should be replaced by the full name of the method 2. The main idea and task are not very clear from the abstract. I suggest adding a sentence with an example of the task you aim to solve somewhere in the beginning of the abstract.

Correctness: Both the theoretical part and experimental analysis seem correct to me.

Clarity: The paper is mostly well-written, although some sentences seem to be too convoluted.

Relation to Prior Work: The prior work section is good, although some papers on multitask learning could also be mentioned.

Reproducibility: Yes

Additional Feedback: Great work!


Review 2

Summary and Contributions: The paper proposes a domain transfer network (conditional invertible neural network) that can translate between fixed representations without having to learn or finetune them. This save a lot of computation and enables the use of very expensive models (BigGAN, BERT) with any further training from scratch. Experiments are performed on diverse conditional image synthesis tasks, such as image-to-image and text-to-image generation.

Strengths: A generic approach that allows to translate between fixed off-the-shelf model representations is proposed, which saves a lot of computation and use of advanced models. The proposed method does not require any gradient computations on the expert models The proposed method achieves state-of-the-art performance on various different domain transfer problem. Results on all tasks are impressive compared to other baselines. The network demonstrates following capabilities: (i) providing generic transfer between diverse domains, (ii) enabling controlled content synthesis by allowing modification in other domains, and (iii) facilitating diagnosis of existing representations by translating them into an easily accessible domain.

Weaknesses: Authors mostly report, FID results in their experiments. For instance, in text-to-image synthesis or image-to-image translation they are not as reliable. One may choose to use R-precision and classification accuracy in addition to FID to check translation accuracy. Would be useful to include FID in Table 1. The choice of StarGAN for image-to-image translation may not be fair as there many advanced models published recently. Line 1-3: Highly confusing sentence to begin the paper.

Correctness: Claims and method seem to be correct. Authors successfully scale the proposed method to many different tasks with their experiments.

Clarity: The paper is easy to understand and well written. Transaction between sections are good and I haven’t experienced many typos.

Relation to Prior Work: Authors successfully discusses other similar works in their related work but their problem differs slightly than the literature. For experiments, mostly state-of-the-art models are used.

Reproducibility: Yes

Additional Feedback: The implementation of the proposed method should be possible following the sup. material. # Post Rebuttal: I find the paper significant. My only concern is the lack of task-dependent evaluation metric: R-precision for text-to-image synthesis, attribute recognition performance for attribute modification task.


Review 3

Summary and Contributions: This paper proposes a generic feature-level domain transfer method that can translate between fixed representations from existing domain-specific expert models without having to learn or finetune them. Such a method enables people to efficiently reuse prior knowledge of heavy/expensive models learned from large scale datasets. Extensive experiments on diverse conditional image synthesis tasks show that the generality of the proposed method. I believe this work can benefit various research communities. ===================== I have read the authors' feedback and other reviews. I keep my inital rating and believe this nice work can benefit many research fields.

Strengths: The proposed method looks simple and efficient. It provides a new way to reuse prior knowledge in the well trained domain-specific expert models, which helps to save time, computation cost, and energy, especially for individual researchers. To enable the representation level transfer, the authors analyze the difficulty that representations of two arbitrary domains are not necessarily isomorphic which implies a non-unique mapping from zΦ to zΘ. Then, the authors propose to use a residual variable v as well as INN to constrain the unique determination of zΘ from v for a given zΦ. Such analysis and design are intuitive and reasonable. The ablation study of replacing cINN with an MLP is also provided in the supplementary material. While I would suggest moving it to the main paper, as it is a good and important experiment to show the necessity of using INN in practice. On some translation tasks, the proposed method achieves comparable SOTAs performance by reusing the existing domain-specific expert models. The paper is easy to follow, although many details are provided in the supplementary material due to space limitations.

Weaknesses: As the proposed method is built on top of the invertible neural network (INN), I would expect some brief introduction and related work discussion on this.

Correctness: Yes

Clarity: Yes

Relation to Prior Work: Yes

Reproducibility: Yes

Additional Feedback: The proposed method is some kind of representation domain transfer, it would be good to discuss the related works and compare the proposed method with them if applicable. Minor issues: Line 159: can be be -> can be Line 190: the right bracket is not matched. The references are not in a good format. Some conference/journal names are missing.


Review 4

Summary and Contributions: To leverage the powerful pretrained models in different specific domains, the authors design a conditional invertible neural network for generic domain transfer between different off-the-shelf model representations. The proposed method shows SOTA performance on several domain transformation problems.

Strengths: - This paper is well-motivated, which aims at leveraging the powerful pretrained model in different domains for cross-domain tasks. It is a critical problem in the machine learning community. This work, apparently, clearly formulates the problem and solves it with a novel method. - The proposed method is a generic approach to solve domain transformation problems. Therefore, it can work as an add-on module to apply on several domains/methods. Correspondingly, experimental results on several problems show the consistent effectiveness of the proposed method, which is convincing and impressive. - The proposed method is novel. It is implemented with a cINN, which can transfer between domains without any alternation. - Comprehensive experiments validate the claims by the authors. The experiment settings are clear and solid. - This paper is well-organized and easy to understand. Mathematical formulation of the problem is reasonable.

Weaknesses: - This job will be more impressive and solid if a dual-way translation problem setup or experiment setting, where invertibility of the cINN is utilized in the inference stage, can be designed.

Correctness: Yes.

Clarity: Yes. The paper is well-organized and clear. The methods and experiments are clearly stated.

Relation to Prior Work: Yes.

Reproducibility: Yes

Additional Feedback:

[Author Response · NeurIPS 2020]

# Network-to-Network Translation with Conditional Invertible Neural Networks

## Author Response

We thank all the reviewers for their positive feedback and for valuing the importance of the problem and the novelty of our approach, and for acknowledging its potential benefits for a variety of research communities as demonstrated by a comprehensive set of experiments.

We will incorporate valuable suggestions regarding related work into the final version and will go into more detail about multi-task learning and INNs in general. Thanks also for the comments on writing style and title, which we will gladly take into account. In particular, as suggested by **Reviewer 1**, we will replace the abbreviation "cINN" in the title by the full name of the method. We furthermore agree with **Reviewer 3** about the importance of the comparison of our cINN with an MLP and will move parts of this section from the supplementary material into the main text.

**Reviewer 2** suggests that the quantitative comparison of our method with the given other works for the *text-to-image translation* task will benefit from additional metrics beside the Inception Score (see Tab. 1 in the main paper). We generally agree with this statement and therefore provide an additional comparison in terms of FID scores:

|  | our | DM-GAN | AttnGAN | Mirror GAN | SD-GAN |
|---|---|---|---|---|---|
| FID $\downarrow$ | **30.63** | 32.64 | 35.49 | no pretrained model | no code available |

To obtain these scores, we used the authors' official implementation available at `https://github.com/MinfengZhu/DM-GAN`. Note, however, that our method utilizes a pretrained expert generator which was trained on the *ImageNet* dataset (i.e. BigGAN, see l.173). Thus, we evaluate FID scores w.r.t. the validation split of the *ImageNet* dataset.

We furthermore agree with **Reviewer 4** that a detailed study of dual-way translation is an interesting avenue for future works.

[Meta-Review · NeurIPS 2020]

All the reviewers agree the work is original and potentially impactful. Two reviewers rate the paper top 15%, one rates top 50%, and one rates a good submission. The work is timely as it provides an efficient way to leverage large-scale networks trained by resourceful institutions for various novel translation applications such as text-to-image or image-to-image. If we can link the latent space learned by a GPT3 and a BigGAN, we could achieve high-quality text to image or image to text translation tasks. The major contribution is a general framework for learning the mapping between two independently learned semantic space using a conditional invertible neural network. After consolidating the reviews and rebuttal, the AC agress with the assessment and congratulate the authors for the acceptance.